# Tailoring Apixaban in Nanostructured Lipid Carrier Enhancing Its Oral Bioavailability and Anticoagulant Activity

**DOI:** 10.3390/pharmaceutics15010080

**Published:** 2022-12-27

**Authors:** Mohamed F. Zaky, Mohamed A. Megahed, Taha M. Hammady, Shadeed Gad, Mamdouh Mostafa Ghorab, Khalid M. El-Say

**Affiliations:** 1Department of Pharmaceutics and Pharmaceutical Technology, Faculty of Pharmacy, Egyptian Russian University, Cairo 11829, Egypt; 2Department of Pharmaceutics and Industrial Pharmacy, Faculty of Pharmacy, Suez Canal University, Ismailia 41522, Egypt; 3Department of Pharmaceutics, Faculty of Pharmacy, King Abdulaziz University, Jeddah 21589, Saudi Arabia

**Keywords:** apixaban, venous thromboembolism, nanostructured lipid carrier, box–behnken design, industrial development, in vivo pharmacokinetic, anticoagulant activity

## Abstract

Apixaban (Apx), an oral anticoagulant drug, is a direct factor Xa inhibitor for the prophylaxis against venous thromboembolism. Apx has limited oral bioavailability and poor water solubility. The goal of this study was to improve the formulation of an Apx-loaded nanostructured lipid carrier (NLC) to increase its bioavailability and effectiveness. As solid lipid, liquid lipid, hydrophilic, and lipophilic stabilizers, stearic acid, oleic acid, Tween 80, and lecithin were used, respectively. Utilizing Box–Behnken design, the effects of three factors on NLC particle size (Y_1_), zeta potential (Y_2_), and entrapment efficiency percent (Y_3_) were examined and optimized. The optimized formula was prepared, characterized, morphologically studied, and pharmacokinetically and pharmacodynamically assessed. The observed responses of the optimized Apx formula were 315.2 nm, −43.4 mV, and 89.84% for Y_1_, Y_2_, and Y_3_, respectively. Electron microscopy revealed the homogenous spherical shape of the NLC particles. The in vivo pharmacokinetic study conducted in male Wistar rats displayed an increase in AUC and C_max_ by 8 and 2.67 folds, respectively, compared to oral Apx suspension. Moreover, the half-life was increased by 1.94 folds, and clearance was diminished by about 8 folds, which makes the NLC formula a promising sustained release system. Interestingly, the pharmacodynamic results displayed the superior effect of the optimized formula over the drug suspension with prolongation in the cuticle bleeding time. Moreover, both prothrombin time and activated partial thromboplastin time are significantly increased. So, incorporating Apx in an NLC formula significantly enhanced its oral bioavailability and pharmacodynamic activity.

## 1. Introduction

Venous thromboembolism (VTE) is considered one of the major causes of mortality and morbidity globally. Deep vein thrombosis (DVT) and pulmonary embolism (PE) account for a significant percentage of the disease load in both outpatient and inpatient settings [1,2]. Until the last ten years, the only options for treating VTE were parenteral drugs, including warfarin, fondaparinux, unfractionated heparin, and low molecular weight heparin. The diversity in medication exposure, narrow therapeutic range, variability of administration methods, regular coagulation monitoring, and multiple drugs and dietary interactions lead to difficulties in their use and, thus, decreased patient adherence to treatment [3]. Furthermore, the accessibility of outpatients is essential because about half of VTE patients are neither hospitalized nor recovering from a serious disease [4]. Many of these drawbacks have been overcome with the introduction of direct-acting oral anticoagulants (DOACs) for managing, and secondary prevention of VTE [5,6,7,8].

Apixaban (Apx), a powerful direct-acting oral anticoagulant medication, is used as a prophylactic therapy to prevent VTE after total knee or hip replacement surgery [9]. It has been marketed by Bristol-Myers Squibb/Pfizer under the trade name Eliquis since its approval by the FDA on 28 December 2012, for the treatment of non-valvular atrial fibrillation in patients at high risk of stroke and systemic embolism [10]. Apx exerts its action by inhibiting both free and clot-bound factor Xa, thus reducing the production of thrombin and the formation of thrombus. Although it doesn’t directly affect platelet aggregation, it indirectly prevents thrombin’s induction of platelet aggregation [11,12]. The dose of Apx for the treatment of VTE is 2.5 mg twice daily up to 5 mg twice daily [13,14]. The C_max_ of Apx after repeated oral administration of 2.5 mg twice daily of the marketed oral tablets (Eliquis^®^) was found to be 62.3 ng/mL with AUC_0-12_ equals 462.8 ng.h/mL and approx. 50% absolute bioavailability. It also reported that The C_max_ of Apx after 7 days repeated oral administration of 25 mg twice daily was found to be 716.6 ng/mL with AUC_0-12_ equals 5850.3 ng.h/mL [12]. Apx has poor water solubility (0.028 mg/mL) and a relatively low oral bioavailability of roughly 50% [15]. The first-pass metabolism of Apx in the gastrointestinal tract (GIT) and liver, as well as incomplete absorption in the gut, are the main causes of the drug’s low oral bioavailability [16]. Because of its poor solubility, increasing the dose higher than 25 mg results in more reduction in oral bioavailability [17]. Additionally, Apx has a significant bleeding side effect, especially after multiple drug administration [18,19]. Therefore, generating new formulations of Apx that increase its solubility and subsequently improve its bioavailability may be interesting and crucial.

Since the development of nanotechnology, it has become possible to incorporate active pharmaceutical ingredients (APIs) into appropriate delivery systems with the necessary physicochemical qualities for deposition through the GI route in a targeted and/or safe way. By improving API solubilization, delivery system mucoadhesion behavior, and cellular absorption of particulate delivery systems, nanotechnology may overcome the main challenges associated with poorly absorbable and/or insoluble drugs [20,21].

Researchers’ interest in lipid-based nanoparticulate delivery methods has grown since they are biocompatible, largely biodegradable, and have numerous biomedical applications [22,23]. This class of nanoparticles has an advantage over other delivery systems due to their nano-sized dimension and lipidic composition. Lipids protect APIs from degradation by enzymatic, chemical, oxidative, and electromagnetic processes [24,25]. Lipid-based nanoparticles are frequently formulated as solid lipid nanoparticles (SLNs) and nanostructured lipid carriers (NLCs). They typically contain a variety of physiological lipids and are surfactant stabilized [26,27]. The original lipid nanoparticle generation, known as SLNs, is solid at both body and ambient temperatures. They produce colloidal suspensions with particles ranging in size from 40 to 1000 nm, frequently with lipids serving as the matrix system. Unfortunately, because of the polymorphic change of the lipids, SLNs experience considerable drug ejection during storage [28]. Moreover, the tightly packed lipid crystal structure, which has few defects and provides a minimal area for API accommodation, gave SLNs a poor loading capacity [29]. Therefore, NLCs, the second generation of lipid nanoparticles, have arisen to address these SLN-related drawbacks. The formulations of NLCs consist of a mixture of solid lipid and liquid oil, in contrast to SLNs. The NLCs keep the solid matrix at both body and room temperature in spite of the liquid oil [30,31]. The solid lipid matrix’s crystallinity is disturbed by the liquid oil, and the ensuing gaps enable the incorporation of higher amounts of API [32]. Increasing drug loading is made possible by the composition and structure of NLCs, which also boost the formulation’s storage stability [33]. Moreover, the use of NLCs was successfully reported to enhance the oral bioavailability of several drugs with poor oral bioavailability. This was confirmed through the marvelous increase in the AUC of the NLC system compared to pure drugs’ suspensions where it was 26.3, 16.5, 8.3, 7.2, and 7.5 folds increase for Nintedanib Esylate [34], lopinavir [35], iloperidone [36], Tacrolimus [37], and docetaxel [38], respectively.

Experimental designs are thought to be useful when developing lipid nanoparticles and other drug delivery systems because they enable the simultaneous examination of many variables in a limited number of experimental runs [39,40]. The Box–Behnken design (BBD), a very effective optimization technique, is one of the experimental designs used for optimization. For the estimation of relationships between both dependent and independent variables, mathematical models are created [41]. This research aims to augment Apx bioavailability and effectiveness with reduced dosing and incidence of side effects to ensure adherence and patient compliance. This approach would be achieved via formulating an optimized Apx-loaded NLC formulation with optimum particle size, zeta potential, and entrapment efficiency that enhances in vitro drug release, in vivo bioavailability, and efficacy of Apx.

## 2. Materials and Methods

### 2.1. Materials

Apixaban was a gift sample from Apex Pharma (Badr city, Egypt). Oleic acid was obtained as a gift from Egyptian International Pharmaceuticals Industries Company (EIPICO) (10th of Ramadan City, Egypt). Lecithin was purchased from Alfa Aesar (Haverhill, MA, USA). Tween 80, stearic acid, and potassium phosphate dibasic anhydrous (K_2_HPO_4_) were purchased from Loba Chemie (Mumbai, India). Orthophosphoric acid was obtained from Biochem (Cairo, Egypt). Sodium lauryl sulphate (SLS) was procured from Research Lab Fine Chemical Industries (Mumbai, India). Chloroform was purchased from Panreac Quimica SA (Barcelona, Spain). Acetonitrile HPLC grade was purchased from Merck (Merck, Darmstadt, Germany). Methyl alcohol absolute, ChromAR^®^ HPLC was obtained from Macron fine chemicals (Center Valley, PA, USA).

### 2.2. Methods

#### 2.2.1. Determination of Variables through a Preliminary Study

To investigate how lipophilic stabilizer, surfactant concentration, and the solid-to-liquid lipid ratio affect NLC formulations, a preliminary study was carried out by preparing different Apx-loaded NLC formulations. Changed weight ratios of Stearic acid (SA), as a solid lipid, to Oleic acid (OA), as a liquid lipid, were used for the development of Apx NLC [42]. Tween 80, at altered amounts, was tested as a hydrophilic surfactant because the physical stability and particle characteristics of NLC depend on the chemical composition and concentration of the emulsifier [43,44]. To test the effect of the addition lipophilic stabilizer, lecithin was incorporated because it can decrease aggregation and stabilize the NLC formulations [45]. Six preparations were prepared, two of them with different lipids’ weight ratios as follows, SA:OA, (3:1) and (1:1). While another two preparations were prepared to test the effect of Tween 80 as a hydrophilic surfactant with two weight ratios concerning lipids as follows, SA:OA:Tween 80, (3:1:1) and (3:1:2). Moreover, the effect of addition of lecithin was investigated through formulating another two preparations were prepared with weight ratio in respect to the other ingredients as follows, SA:OA:Tween 80:lecithin, (1:1:1:1) vs. the same ratios without the addition of lecithin. The study was performed to test the effect of the aforementioned factors on particle size, zeta potential, and the entrapment efficiency of the developed Apx-loaded NLC formulae. This study was conducted to determine the factors and the levels that will be adopted in the optimization study.

#### 2.2.2. Experimental Design

Box–Behnken design (BBD) was employed to optimize Apx-loaded NLCs [39]. Amount of SA (X_1_), Tween 80 (X_2_), and lecithin (X_3_) were chosen as independent variables to study their effects on particle size (Y_1_), zeta potential (Y_2_), and entrapment efficiency (Y_3_). The optimization goal was to minimize Y_1_ and maximize Y_2_ and Y_3_. Using the Statgraphics^®^ Centurion XV software, version 15.2.05 (StatPoint, Inc., Warrenton, VA, USA), the mathematical relationships between the observed responses and the designated factors were explained as polynomial equations, and ANOVA determined the significance of these relationships. Both dependent and independent variables and their levels are presented in Table 1.

#### 2.2.3. Preparation of Apx-Loaded NLCs

With some modifications, a thin film hydration-ultrasonication method was applied to formulate both Apx-loaded and drug-free NLCs [46,47]. According to the levels examined, the required amounts of SA, OA, with or without lecithin, were weighed. A constant weight of Apx (10 mg) was also added to the mixture. The mixture was added to the pear-shaped flask of the Büchi-M/HB-140 rotary evaporator (Flawil, St. Gallen, Switzerland), and dissolved completely using sonication in a chloroform–methanol mixture at a 2:1 ratio. Under vacuum, in a water bath at 60 °C for 30 min, organic solvents were slowly and thoroughly evaporated, leaving a thin, dry lipid film deposited on the flask wall. The aqueous phase was formed by dissolving the necessary quantity of Tween 80 in 20 mL of deionized water. This solution was used to hydrate the thin film at a 60 °C water bath for 60 min in the presence of 7–10 glass beads with a diameter of 4 mm to ensure complete hydration of the dry lipid thin film. The dispersion was then sonicated using a probe sonicator, Vibra cell VCX 750 (Sonic and Materials Inc, Newtown, CT, USA) for 4 min at 40% amplitude and kept at −20 °C until further investigations [48].

#### 2.2.4. Separation and Washing of Apx-Loaded NLCs

The frozen Apx NLCs dispersion was thawed above the preparation temperature (65 °C) since freeze-thawing is thought to increase the entrapment efficiency significantly [41]). Using a High-Speed Refrigerated Centrifuge Sigma 3-30 KS (SIGMA Laborzentrifugen GmbH, Osterode, Germany) at a temperature of approximately −4 °C and a force of 20,000 g for 120 min, free Apx was separated from the NLC dispersion. Using a vortex mixer and re-dispersion in deionized water, the NLC particles were washed before being centrifuged. This washing procedure was carried out twice to ensure that the medication was no longer present in the spaces between the NLC vesicles [49].

#### 2.2.5. Characterization of Apx-Loaded NLCs

##### Determination of the Particle Size and Zeta Potential

Before analysis, all formulations were diluted with deionized water to an appropriate dilution (1/10) and then sonicated for five minutes to remove air and break up any particle clumps [50]. Then, the mean particle diameter expressed in nm, surface charge expressed in mV, and polydispersity index (PDI) for all generated Apx-loaded NLC formulations were determined at 25 °C using quasi-elastic light scattering (QELS) based on laser diffraction (NICOMP^TM^ 380 ZLS, Santa Barbara, CA, USA). The mean of three cycle assessments of the mean particle diameter served as the value for particle diameter and zeta potential.

##### Determination of the Entrapment Efficiency (EE%)

Encapsulation efficiency was directly determined by accurately withdrawing 1 mL of the previously mentioned washed NLCs dispersion using a 100–1000 µL micropipette from Dragon Lab Scientific Co., Ltd. (Beijing, China). The particles were then burst with addition of methanol, releasing the drug that had been contained, and the volume was completed to 10 mL. Finally, sonication was applied to obtain a clear solution, which was then spectrophotometrically analyzed using a UV–visible spectrophotometer, Jasco V-630 (Tokyo, Japan) at a wavelength of 279 nm [51]. As a control in the measurement, drug-free NLCs were also treated using the same method. Following measurements thrice, Equation (1) was used to compute the EE% as follows [52,53]:(1)$$\mathrm{EE}\%=\frac{\mathrm{Amount}\;\mathrm{of}\;\mathrm{entrapped}\;\mathrm{Apx}}{\mathrm{Total}\;\mathrm{amount}\;\mathrm{of}\;\mathrm{Apx}\;\mathrm{added}\;\mathrm{in}\;\mathrm{the}\;\mathrm{formulation}}$$

#### 2.2.6. Prediction, Preparation, and Characterization of the Optimized Formula

BBD was applied effectively, and experiments were created by choosing the input variables with the selected levels. Using Statgraphics software, the response surface methodology (RSM) developed for responses demonstrated each input parameter’s effect and how it interacted with other parameters to forecast and achieve the optimum Apx NLC formula. Particle size, zeta potential, and EE% measurements were then used to assess the optimized formula. 

##### In Vitro Apx Release from Optimized NLC Formula

The dialysis bag diffusion method was used to compare the in vitro release characteristics of Apx from the optimized NLC formula to pure medication [54]. A dialysis bag (VISKING^®^ Dialysis Tubing MWCO12,000–14,000) with dimensions of 4 cm in length and 2.1 cm in diameter was filled with a fixed volume (3 mL). To ensure optimal diffusion, the bag was immersed in the release medium the day before the experiment for one night. The dialysis bags were submerged in 50 mL phosphate buffer solution (PBS) with 0.05% SLS to create the sink condition and increase Apx solubility [55]. The solution’s pH was then raised to 6.8 with the use of orthophosphoric acid and dipotassium hydrogen phosphate. The release was carried out on a benchtop ThermoStableTM IS-20 incubator shaker (Daihan Scientific Co., Ltd., Wonju, Republic of Korea). The shaker was adjusted to 140 rpm and maintained at 37 °C. One milliliter samples were taken at predefined intervals of 0.5, 1, 2, 4, 6, 8, and 12 h and promptly replaced with new medium.

The amount of the drug released in the media was determined through UV-spectrophotometric analysis of the withdrawn sample at a wavelength of 279 nm. The experiments were carried out three times independently, and the results were represented as the cumulative drug release percentage over a 24 h period.

##### Mathematical Modeling of Apx In Vitro Release from Optimized NLC Formula

By comparing the correlation coefficients (r), the kinetic release models (zero, first, second-order, Hixon-Crowell release, Baker-Lonsdale, and Higuchi diffusion) were mathematically matched to the release study data, with the model with the greatest coefficient being chosen as the best fit.

##### Transmission Electron Microscopy

To study the morphology and correctly measure the size of the developed nano-formulation, a transmission electron microscope Jeol Jem-2100 (Tokyo, Japan) was employed. Deionized water considerably dilutes the optimized formula to an appropriate intensity, allowing for a good vision of the generated NLCs. On a grid covered in carbon, one drop of the diluted NLCs suspension was applied, and it was then left for one minute to allow some of the particles to adhere to the carbon substrate. The surplus dispersion was taken out using a piece of filter paper. Filter paper was used once again to remove any remaining staining solution after adding a drop of 1% phosphotungstic acid solution as a staining solution. The sample was then examined using an electron microscope after being allowed to dry in the open air [56,57].

#### 2.2.7. In Vivo Evaluation of Optimized Apx-NLC Formula

##### Protocol and Animal Preparation

Animal studies were carried out in compliance with the protocol (#202104PHDA1) approved by the research ethics committee of the Faculty of Pharmacy, Suez Canal University. The studies were performed on male Wistar rats with a mean body weight of 200 ± 20 g. During the study, the animals were housed in plastic mesh cages with water and regular laboratory feed under normal lighting (12 h light/dark cycles), humidity, and temperature conditions. All rats were denied food and drink overnight the day before the experiment. The rats were randomized into three groups; Group A (*n* = 6) was administered normal saline (control group), Group B (*n* = 12) was given a single oral dose of (60 mg/kg) of a free drug suspension and Group C (*n* = 12) which was administered the optimized Apx-loaded NLC of the same dose [58]. Free Apx suspension was prepared in deionized water using glycerin and 0.2% gum tragacanth [39]. Afterward, the groups were further subdivided to perform pharmacokinetic and pharmacodynamic studies. Statistical analysis of results was done using an unpaired Student *t*-test to compare the results of groups B and C to each other and to control results (group A). Data were presented as the mean ± SD with a *p*-value of less than 0.05 was considered significant.

##### Chromatographic Conditions

A reverse phase (RP) high-performance liquid chromatography (HPLC) (Waters Alliance e2695, Milford, MA, USA) system with an Agilent ZORBAX Eclipse C18 column (250 *×* 4.6 mm, 5 µm) (Agilent, CA, USA) and a PDA detector, was employed for the analysis of Apx plasma concentrations in experimental animals. The mobile phase consisted of a buffer solution (0.1% phosphoric acid in water): methanol in 50:50 (*v*/*v*) with a flow rate of 1.0 mL/min and a 100 µL injection volume, and peaks were detected at 280 nm. The column temperature was set at 30 °C, and the total run time was 6 min [59,60].

##### Pharmacokinetics Study

Six rats from each group (B and C) were chosen to perform the pharmacokinetic study. At predetermined intervals (0.25, 0.5, 1, 2, 3, 4, 6, 8, 12, and 24 h), arterial blood samples were drawn into a 1/10 volume of 3.8% Na-citrate and centrifuged for 5 min at 10,000× *g* using a Microfuge E centrifuge, Beckman Instruments, (Palo Alto, CA, USA) to obtain plasma, and then frozen at −80 °C using ultra-low temperature freezer WUF-25, Daihan Scientific Co., Ltd., (Seoul, Republic of Korea) till analysis. Afterward, the obtained plasma (100 µL) was mixed with 50 µL of rivaroxaban (0.010 µg/mL, used as the internal standard, IS) and 500 µL of acetonitrile. After centrifugation (20 min at 5000 rpm), 100 µL of supernatant was injected into HPLC for analysis. The add-on program, PK Solver 2.0 software, was used to calculate the different pharmacokinetic parameters expressed as the mean ± SD.

##### Pharmacodynamics Study

Cuticle Bleeding Time (CBT)

Nine rats (3 from each study group) were chosen to perform the CBT test. A single-edged razor blade was used to cut the hind paw toenail where the nail meets the quick. The cuticle was instantly submerged in 37 °C Ringer’s solution. The time (in seconds) until bleeding ceased without rebleeding for 30 s was calculated while observing under binocular 3× magnification. The period beyond which the operator exerted external pressure to stop the bleeding was considered maximal bleeding [61].

Prothrombin Time (PT) and Activated Partial Thromboplastin Time (APTT)

To determine the impact of the Apx-loaded NLC on the coagulation process in comparison to the free drug suspension and control group, clotting time parameters, including PT and APTT, were assessed [62,63]. A fraction of the rats (*n* = 3 for each study group) had their blood samples taken two hours after delivery, and they were then treated, as mentioned before under the pharmacokinetics section, to produce plasma, which was used right away to evaluate PT and APTT [64]. Baxter Healthcare Corporation (Deerfield, IL, USA) provided the Dade Thromboplastin-C reagent, which was used to assess PT utilizing a mechanical clot detection technique with an ISI of 2.0. APTT was again determined using Dade Actin FSL reagent, Siemens Healthineers (Erlangen, Germany), applying mechanical clot detection technique.

#### 2.2.8. Statistical Analysis

All statistical analyses were performed using GraphPad Prism 8 Software (GraphPad Inc., La Jolla, CA, USA). A two-way ANOVA with Sidak’s multiple comparisons test was done to determine various groups’ significance. Results with *p* < 0.05 were considered significant.

## 3. Results and Discussion

### 3.1. Preliminary Study

#### 3.1.1. Variables Affecting Entrapment Efficiency of Apx-Loaded NLCs

The results of the preliminary study demonstrated that Apx-loaded NLC containing the lipophilic stabilizing agent, lecithin, exhibited significantly higher entrapment efficiency of 60.6 ± 1.56% than those not lecithin with EE of 52 ± 1.63% at the weight ratios investigated (*p*-value < 0.05). The higher entrapment efficiency can be explained by the fact that lecithin addition lowers the likelihood of drug loss to the external phase and increases the space available for drug incorporation by generating several layers around the particle, thus improving entrapment efficiency. These results are in concordance with a previous study that investigated the effect of lecithin on the entrapment efficiency of NLC systems [65].

The second variable that affected the EE of Apx-loaded NLCs was the solid lipid (stearic acid) to liquid lipid (oleic acid) ratio; Apx NLC formulae with decreased SA: OA ratio (1:1) (i.e., higher liquid lipid ratio) showed higher EE of 44.3 ± 0.98% than the corresponding formula with higher SA:OA ratio (3:1) which showed EE of 40.2 ± 1.07% (*p* < 0.05). These results were in good agreement with previously reported studies which showed that the presence of a liquid phase gives the system’s porous network adequate space to accept more drug particles, improving drug entrapment efficiency [66]. Additionally, it was noted that adding liquid lipids to solid lipids could significantly disturb the crystal order. The resulting matrix of lipid particles shows significant flaws in the crystal lattice. It provides more space for drug molecules, improving the ability to load drugs and the percentage of drug entrapped [67].

It was found that increasing the Tween 80 (hydrophilic surfactant) amount leads to an increase in the entrapment efficiency; however, this increase was insignificant (*p >* 0.05). The insignificant increase in EE% was in agreement with that of Kheradmandnia et al. [68], who reported that increasing the surfactant concentration from 0.5 to 1.5% led to no discernible difference in entrapment efficiency.

#### 3.1.2. Variables Affecting the Particle Size and Zeta Potential of Apx-Loaded NLCs

The preliminary study’s findings showed that the addition of lecithin reduced particle size from 261.7 nm to 198.7 nm, and an increased zeta potential from −26.5 mV to −41.3 mV. The interfacial tension between the lipid and aqueous phases may have decreased, causing the generation of smaller-sized emulsion droplets and the decrease in particle size [69]. By creating a steric barrier on the surface of the particles, lecithin effectively stabilized the particles, protecting smaller particles and preventing their coalescence into larger ones [70]. The increase in the magnitude of zeta potential towards the negative side may be caused by the phospholipid composition of lecithin, which gives the surfactant a net large negative charge [68].

Another variable influencing particle size and zeta potential was the SA: OA ratio. Increasing the SA ratio relative to OA leads to an increase in particle size from 280.8 nm to 443.6 nm and decreasing zeta potential from −23.7 mV to −19.5 mV. It was found that adding more liquid lipids (OA) to formulations reduced the surface tension and viscosity of NLC suspensions, resulting in the formation of smaller, more uniform surface particles [67]. Because oleic acid contains a negative charge due to its carboxylic group, which enhances the negativity of the NLC particles generated, the phenomenon for zeta potential may be explained. As a result, lowering the OA content relative to SA will lower the zeta potential [71].

Finally, it was observed that Apx-loaded NLCs prepared using a higher amount of Tween 80 showed decreased particle size from 443.6 nm to 368.3 nm and decreased zeta potential from −19.5 mV to −15.8 mV. The decrease in particle size with increasing tween concentration can be attributed to the availability of additional surfactant to cover the water-lipid interfaces created during NLC preparation. This additional surfactant adsorbs rapidly to particle surfaces and lower interfacial tension between the water and lipid phases, stabilizing newly formed surfaces and creating smaller particles [72,73]. The fact that Tween 80 is a nonionic surfactant and that increasing its concentration will cause it to be adsorbed on the surface of particles, thus reducing the net charge at the particle surface, explains why the magnitude of zeta potential decreases when Tween 80 concentration is increased [74].

### 3.2. Optimization of Apx-Loaded NLCs

#### 3.2.1. Estimation of the Quantitative Impacts of the Factors

The compositions and observed response values of 15 experimental runs, including triplicate center points, are shown in Table 2. Multiple regression analysis using Statgraphics software together with a two-way ANOVA was employed for the statistical examination of BBD equations. The estimated factor effects and *p*-values for the three ANOVA-generated results are shown in Table 3. Figure 1 displays the major effect plots of the factors on the examined responses. The effect is deemed significant if a factor effect deviates from zero and the *p*-value is less than 0.05. A positive sign refers to a synergistic effect of a factor, i.e., a direct relationship between the effect of the factor and the tested response

As seen in Table 3, the antagonistic impact of any component, or an inverse relationship between the effect and the response, is denoted by a negative sign. The Pareto charts of the relationships between various factors and the responses along with their major ones are also shown in Figure 2. Additionally, the impact of all factors on the different responses across the examined factor levels is shown as a 3D response surface plot (Figure 3). It was clear that SA amount (X_1_) has a significant synergistic effect on the particle size (Y_1_) with a *p*-value of 0.0094 while having a significant antagonistic effect on Y_2_ (zeta potential) and Y_3_ (entrapment efficiency) with *p*-values of 0.007 and 0.0015, respectively. The Tween 80 amount (X_2_) had a significant antagonistic effect on the particle size (Y_1_) and zeta potential (Y_2_) with a *p*-value of 0.0169 and 0.0021, respectively.

On the other hand, X_2_ has an insignificant synergistic effect on the entrapment efficiency (Y_3_) with *p*-values of 0.0755. The addition of lecithin (X_3_) showed a significant synergistic effect on zeta potential (Y_2_) and entrapment efficiency (Y_3_), with *p*-values of 0.0001 for both of them. Lecithin, on the other hand, exhibited a significant antagonistic effect with a *p*-value of 0.0046 on Y_1_ (particle size). Additionally, it was evident that the quadratic term of X_1_ had a substantial antagonistic effect (*p*-value of 0.0052) on the Y_1_ and a significant synergistic effect (*p*-value of 0.018) on the zeta potential (Y_2_). The quadratic term of X_2_ demonstrated a significant antagonistic influence on zeta potential with a *p*-value of 0.0077. Regarding the quadratic term of X_3_, it exhibited a substantial antagonistic influence on zeta potential as well as EE% with *p*-values of 0.0002 and 0.0012, respectively, while having a synergistic effect on particle size with a *p*-value of 0.0152. Regarding the interaction terms X_1_X_2_ and X_1_X_3_, respectively, it was discovered that they had a substantial synergistic influence on the Y_1_ with *p*-values of 0.0286 and 0.0074, respectively.

#### 3.2.2. Effects on the Mean Particle Size (Y_1_)

Indicating a consistent distribution of NLC dispersions, the mean measured particle size for all NLC formulae (Y_1_) varied from 188.9 nm for F8 to 543.7 nm for F14, with PDI for all formulations ranging from 0.35 for F8 to 0.57 for F3. The outcomes were very evident, as shown in Figure 1 and Figure 2, that the amount of lecithin (X_3_) was the primary component causing the change in particle size of Apx-loaded NLCs. Lecithin addition was found to have a substantial inverse relationship with the Y_1_. The decrease in particle size from 527.6 nm for F2 to 188.9 nm for F8, with increasing the amount of lecithin from zero to 100 mg while maintaining the levels of X_1_ and X_2_, is a good example of how lecithin affects particle size (Table 2). We observed the same finding in the particle size of F14 and F13, where increasing the lecithin amount from 0 to 100 mg with keeping the levels of X_1_ and X_2_ constant led to a decrease in the particle size from 543.7 nm to 405.1 nm, respectively. This result was consistent with the outcomes of earlier research on the impact of lecithin addition on the average particle size of NLCs [69,70].

Contradictorily, it was found that the SA amount (X_1_) had a significant direct effect (synergistic) on the mean particle size. This was evident when contrasting the particle sizes of F7 and F6, where the increase in SA from 100 mg to 300 mg (i.e., the rise in the SA: OA ratio from 1:1 to 3:1) at the same levels of X_2_ and X_3_ resulted to an increase in particle size from 315.3 nm to 506.9 nm (Table 2). The impact of increasing OA as a liquid lipid on lowering surface tension and reducing the particle size of NLC particles may account for these results [67].

The difference in particle size between F11 and F12 (Table 2), where the amount of Tween 80 was increased from 100 mg to 300 mg at equal levels of X_1_ and X_3_, demonstrated that X_2_ (the amount of Tween 80) had an antagonistic effect on Y_1_. This was evidenced by the decrease in particle size from 515.3 to 493.9, respectively, when the amount of Tween 80 was increased from 100 mg to 300 mg. This could be because Tween 80 can rapidly adsorb on the particle surface, causing a reduction in the interfacial tension between the water and lipid phases, thus producing smaller particles [72,73].

#### 3.2.3. Effects on Zeta Potential (Y_2_)

Zeta potential for all NLC formulae ranged from −23.4 mV for F3 to −47.5 mV for F13. It was clear that lecithin amount (X_3_) is the main factor affecting the zeta potential of Apx-loaded NLCs, as displayed in Figure 1 and Figure 2. The effect of lecithin amount on zeta potential was synergistic, as evidenced by increasing the zeta potential from −25.9 mV to −47.5 mV for F14 and F13, respectively, with the increase in lecithin amount from 0 to 100 mg, at the same levels of X_1_ and X_2_. It should be mentioned that the increase in zeta potential depends on the value, while the negative sign is neglected. Another example of the effect of lecithin on the zeta potential was observed in F2 and F8, where zeta potential increased from −24 mV to −46.8 mV, respectively. Increasing the amount of lecithin may raise the negative charge on the surface of particles since lecithin is a phospholipid and has a net negative charge [68].

On the other hand, it was found that the zeta potential was significantly adversely affected by the SA quantity (X_1_) (Figure 1, Figure 2 and Figure 3). This can be seen from the difference in zeta potential between F1 and F15, where increasing the amount of SA from 100 mg to 300 mg resulted in a drop in zeta potential from −41.2 mV to −38.8 mV at the same level of X_2_ and X_3_. The potential was raised for F15 and F7 from −38.8 mV to −44.6 mV, respectively, by reducing the amount of Tween 80 from 300 mg to 100 mg at the same level of SA and lecithin (Table 2). The same effect was seen for X_2_. This could be explained by the nonionic nature of Tween 80, which, when adsorbs on the surface of the NLC particles, reduces the net charge on the particle surface [74].

#### 3.2.4. Effects on the Entrapment Efficiency (Y_3_)

The range of the entrapment efficiency (Y_3_) for all NLC formulations was from 42.5% to 94.1%; this variation indicates the good choice of the factors and their levels. It was found that factors X_1_ and X_3_ significantly affect the EE%, while an insignificant effect was observed for factor X_2_. Factor X_1_ has an antagonistic impact on Y_3,_ while factor X_3_ has a synergistic one, as shown in Figure 1 and Figure 2. As demonstrated in Figure 1, Figure 2 and Figure 3, it was clear that the leading factor affecting the EE% of Apx-loaded NLC formulae is lecithin amount (X_3_). An increase in lecithin from zero to 100 mg, while not changing the levels of X_1_ and X_2_, increased the entrapment efficiency from 57.2% to 94.1% for F14 and F13, respectively. The EE% decreased from 82.7% for F11 to 48.2% for F9, decreasing the lecithin amount from 100 mg to 0 mg (Table 2). This finding was in agreement with what was stated by Moghddam et al., who found that the addition of lecithin forms multilayers around the particle and leads to a reduction in the amount of drug lost to the external phase and thus improves entrapment efficiency [65].

Regarding X_1_ (SA amount), it was found to have an inverse effect on the EE%. As the SA amount increased from 100 mg to 300 mg in F13 and F8, the percentage of entrapped Apx decreased from 94.1% to 76.5%. This finding was also revealed in F1 and F15, where increasing the SA amount by 3 folds while keeping the levels of X_2_ and X_3_ resulted in a reduction of EE% from 79.4% to 72.4%, respectively. The explanation of these findings could be that the increase in liquid lipid (OA) amount, relative to SA amount, provides enough space to incorporate the drug particles in a system network, thus increasing the entrapped amount [66].

#### 3.2.5. Statistical Analysis and Mathematical Modeling of the Experimental Data

Results of the mean particle size of NLCs (Y_1_), zeta potential (Y_2_), and entrapment efficiency (Y_3_) were statistically tested to generate a mathematical model for each of them. Equations (2)–(4) show the results of the multiple regression analysis for every response variable generated from the best-suited model.
Mean vesicle size (Y_1_) = 246.93 + 2.80 X_1_ − 0.60 X_2_ − 0.39 X_3_ − 0.01 X_1_^2^ + 0.003 X_1_X_2_ − 0.01 X_1_X_3_ − 0.0004 X_2_^2^ + 0.0003 X_2_X_3_ + 0.015 X_3_^2^(2)
Zeta potential (Y_2_) = 29.23 − 0.04 X_1_ + 0.03 X_2_ + 0.46 X_3_ + 0.00008 X_1_^2^ − 0.00001 X_1_X_2_ + 0.00006 X_1_X_3_ − 0.0001 X_2_^2^ + 0.00006 X_2_X_3_ − 0.003 X_3_^2^
(3)
Entrapment efficiency (Y_3_) = 48.73 − 0.04 X_1_ + 0.09 X_2_ + 0.75 X_3_ + 0.00005 X_1_^2^ − 0.0001 X_1_X_2_ − 0.0001 X_1_X_3_ − 0.0001 X_2_^2^ − 0.0002 X_2_X_3_ − 0.003 X_3_^2^(4)

### 3.3. Preparation of the Optimized Apx-Loaded NLC Formula

BBD is used for optimization based on a 3-factor 3-level design with experimental points intermediate from the central point and located on a hypersphere [75]. It helped reach an optimized Apx-loaded NLC formula that met our requirements of smallest particle size, highest zeta potential, and EE%. The final optimized parameters were considered and assessed to find a compromise between various responses to achieve a combination of factor levels that maximize the desirability function. By creating a new formula in accordance with the anticipated model and evaluating it for the responses, as shown in Table 4, the validity of the BBD results was confirmed. The optimized Apx-NLC formula was developed utilizing the acquired optimum values of the tested variables: 100 mg, 299.9 mg, and 75.23 mg of X_1_, X_2,_ and X_3_, respectively. The thin film hydration method was used to prepare the Apx-loaded NLC optimized formula with the aforementioned optimum variables levels, and then it was characterized as mentioned before.

### 3.4. Characterization of the Optimized Apx-Loaded NLC Formula

#### 3.4.1. Particle Size, Zeta Potential, and EE%

Table 4 shows that, by comparing the observed values of responses to the predicted ones, there were no considerable residuals and the predicted error percentage of 2.3%. The values of observed responses were 315.2 nm, −43.4mV, and 89.84% for Y_1_, Y_2_, and Y_3_, respectively, while those of predicted responses were 309.32 nm, −44.43 mV, and 88.27, respectively. These findings along with the low error percentage, indicate the reasonable value of the working design for optimizing Apx-loaded NLCs.

#### 3.4.2. In Vitro Apx Release from Optimized NLC Formula

The in vitro release of Apx from the NLC formula compared to free Apx was tested for 24 h in simulated intestinal fluid (pH 6.8) and is demonstrated in Figure 4a. The graph shows that the release of Apx from the NLC formula was biphasic, with the first phase being a quick release and the second one being a sustained release. The slower later release came from Apx was integrated into the NLC core and released in a sustained pattern through the matrix erosion or degradation. In contrast, the faster initial release came from Apx, rapidly diffusing from the surface [76,77]. Compared to the free Apx, it was clear that the optimized formula showed a slower release. The free Apx released almost 100% after four hours, while a much smaller amount of the drug was released from the formula within the same period.

#### 3.4.3. Mathematical Modeling of Apx In Vitro Release from Optimized NLC Formula

The data from the Apx release was subjected to various kinetic models. Apx release failed to properly fit zero, first or second orders, with r values of 0.794, −0.836, and 0.859, respectively. Moreover, the optimized formula release did not show proper fitting for Baker-Lonsdale nor Hixon-Crowell models with r values of 0.843 and 0.824, respectively. With the highest r value of 0.912, the best-fitting model was the Higuchi diffusion model, successfully describing the kinetics and the mechanism of Apx release from the formula. Knowing that this model best fits the systems that demonstrate release by diffusion across the lipid matrix, it is likely that the primary mechanism of the Apx release was diffusion [78]. This was entirely consistent with the results of earlier research on the kinetics of drug release from NLC formulations [79,80].

#### 3.4.4. Transmission Electron Microscopy

Using TEM, the morphology of the optimized Apx-loaded NLC formula was studied, and the TEM photograph is displayed in Figure 4b,c. The image demonstrates that the Apx-NLC formula nanoparticles were uniformly spherical. The vesicle seems smaller in size when measured using the TEM technique than when measured using the QELS technique. This could be due to the drying steps required in the TEM sample preparation, which leads to the nanoparticle’s shrinkage, while the QELS technique involves hydration steps that keep the nanoparticles’ size [81].

### 3.5. In Vivo Characterization of Optimized Apx-NLC Formula

#### 3.5.1. Pharmacokinetic Study

The plasma concentration-time curve profiles of optimized Apx-loaded NLC formula and free Apx suspension after oral administration to Wistar rats are shown in Figure 5, and the corresponding pharmacokinetic parameters are summarized in Table 5 and displayed in Figure 6. Following oral administration of Apx-suspension, the mean area under the curve (AUC_0-inf_) was found to be (5.753 ± 1.353 μg/mL·h), and the peak plasma concentration (C_max_) was (0.839 ± 0.069 μg/mL). On the other hand, the Apx-NLC resulted in AUC_0-inf_ of (59.38 ± 13.2 μg/mL·h) and C_max_ of (2.24 ± 0.247 μg/mL) with a significant increase by 8 folds (*p*-value < 0.0001) and 2.67 folds (*p*-value <0.0001), respectively, (Figure 6). These results indicate the improvement of oral bioavailability of Apx after administration in the NLC formula. This could be explained by the following suggested mechanisms. First, the nanosized particles have a high surface area thus they are absorbed through the lymphatic system, bypassing the first pass effect and eventually increasing the bioavailability [34]. Moreover, it was reported that Tween 80 acts as a P-gp inhibitor, thus it can increase the transport of NLC through the intestinal mucosa and improves the systemic accessibility of NLCs [82]. Moreover, the non-ionic nature of Tween 80 could reduce the uptake of NLCs by the reticuloendothelial system [36]. Additionally, surfactants act as cell permeability enhancers and have the ability to increase membrane fluidity, thus they could increase intestinal permeability [83]. Finally, the use of lipids with long-chain fatty acids, such as stearic acid, increases the formulation’s lymphatic uptake [84].

Moreover, the half-life (T_1/2_) of Apx-NLC was increased significantly by 1.94 folds (*p*-value = 0.0002), and the clearance (Cl_T_) was significantly decreased by about 8 folds (*p*-value < 0.0001) compared to Apx-susp. The prolongation of T_1/2_ and decrease in Cl_T_ indicate the sustained effect of the prepared NLC formula and can be explained by the fact that NLC can adhere effectively to the gut wall. Thus residence time is prolonged, and consequently, the amount of drug available for clearance is decreased [85,86]. To conclude, the use of NLCs significantly enhanced the oral bioavailability of Apx and that was in good agreement with previously mentioned studies for the effect of NLCs on several drugs as mentioned in Section 1. This could help to decrease the dosing frequency and improve patient compliance and adherence to the therapy, but more clinical investigations are needed.

#### 3.5.2. Pharmacodynamic Study

In order to assess the pharmacodynamic activity of the optimized Apx NLC formula compared to the free drug and control group, bleeding and clotting time assays are carried out. Cuticle bleeding time (CBT) was used as a measure of hemostasis [87], while prothrombin time (PT) and activated partial thromboplastin time (APTT) were used to assess the function of the extrinsic and intrinsic clotting pathways, respectively [88]. As displayed in Figure 7, Apx-loaded NLC produced a significant increase (*p* < 0.05), in all tests relative to control as well as free drug groups. Mean CBT in the control group, free drug suspension group, and Apx-loaded NLC group were 157 ± 15, 296 ± 27, and 470 *±* 28 s, respectively. It is observed that Apx-loaded NLC significantly prolonged bleeding more than the control or free Apx groups, with a *p*-value of 0.0015 and <0.0001, respectively. Apx-loaded NLC increased the prothrombin time by 3.69 ± 0.35 times control (*p* < 0.0001) and by 1.72 ± 0.22 times free Apx suspension (*p* = 0.0006). Regarding APTT, it was found that the mean thromboplastin time for the Apx-NLC formula was 52.03 ± 2.33, while the result of the control group and free drug group was 24.06 ± 2.29 and 38.49 ± 2.4, respectively. This indicates the significant prolongation of APTT caused by the drug NLC formula compared to the other two groups. It was noticed that the impact of Apx on PT was greater than on APTT and this finding is in concordance with previously published data [89]. The pharmacodynamic results are correlated with the pharmacokinetic results which demonstrated the enhanced bioavailability of Apx-loaded NLCs and consequently the efficacy. Overall, the pharmacodynamic results indicate the superiority of the prepared Apx-loaded NLC formula over control and free Apx suspension.

## 4. Conclusions

NLC, as a new generation of lipid nanoparticles, showed a successful enhancement in the oral bioavailability of poorly soluble drugs. An Apx-NLC formula was predicted and optimized using BBD to enhance the bioavailability of Apx as well as the therapeutic efficacy. The optimized Apx-NLC formulation showed optimum particle size, zeta potential, EE%, and sustained in vitro drug release. Finally, the pharmacokinetic study demonstrated the enhancement in oral bioavailability of the Apx NLC. The enhancement confirmed this in the pharmacodynamic results through the assessment of CBT, PT, and APTT. Finally, Apx-loaded NLC is a successful strategy to enhance oral bioavailability as well as the therapeutic efficacy of Apx. Future studies are recommended to confirm the prolonged action and reduced dosing frequency of Apx-loaded NLCs.

## Figures and Tables

**Figure 1 pharmaceutics-15-00080-f001:**
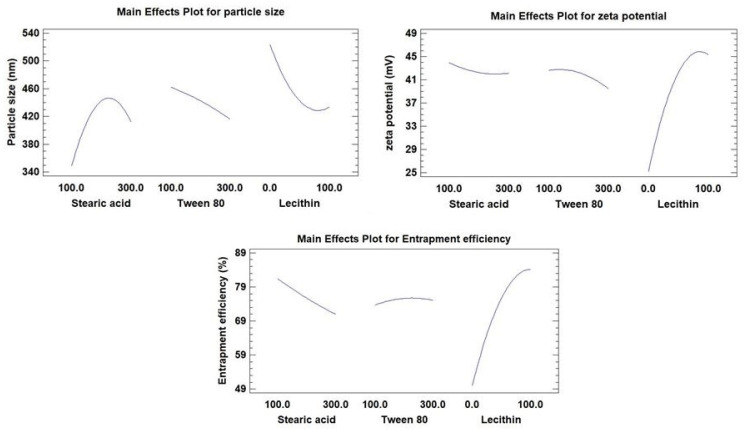
Main effect plots showing the effects of the investigated factors on all responses (Y_1_–Y_3_).

**Figure 2 pharmaceutics-15-00080-f002:**
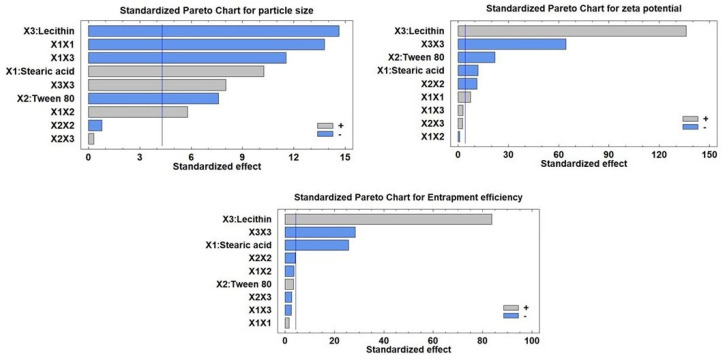
Standardized Pareto charts showing the effects of the investigated factors on all responses (Y_1_–Y_3_).

**Figure 3 pharmaceutics-15-00080-f003:**
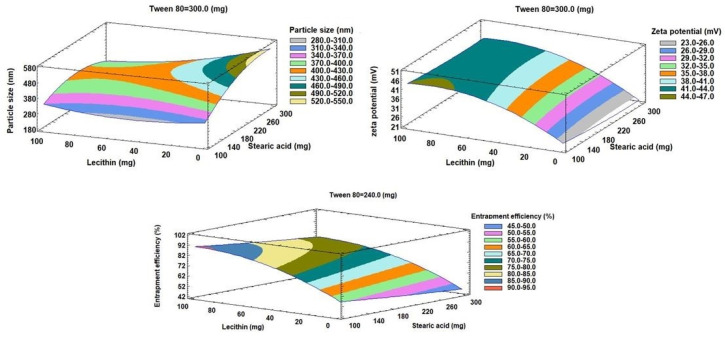
Estimated response surface plots for the effects of the investigated factors on all responses (Y_1_–Y_3_).

**Figure 4 pharmaceutics-15-00080-f004:**
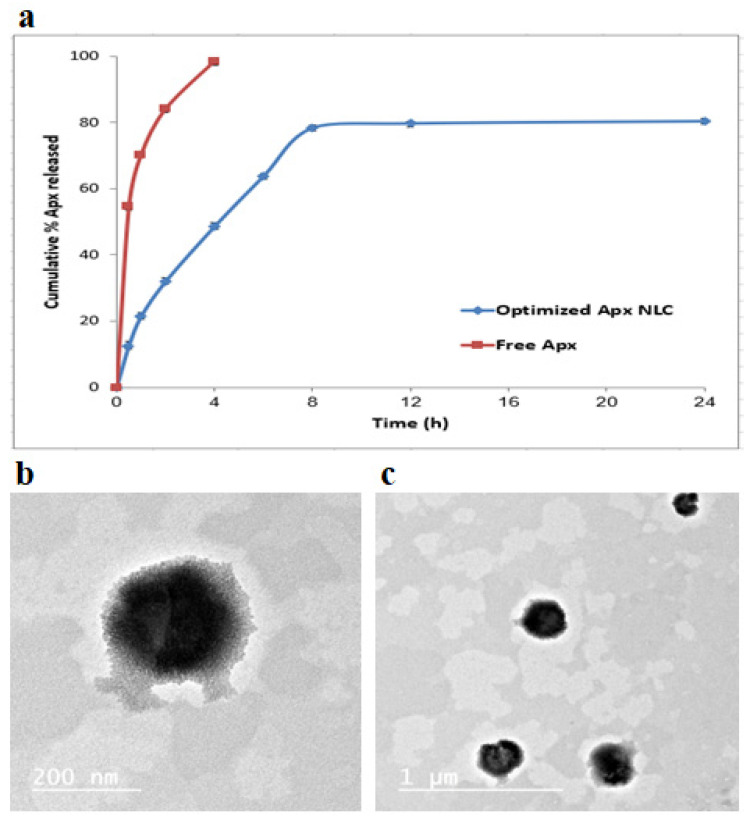
In vitro release profiles for the optimized formulation compared with the free drug in suspension (mean ± SD, *n* = 3) (**a**); photomicrograph of the optimized Apx-NLCs formulation by TEM images (**b**,**c**) at different magnification power.

**Figure 5 pharmaceutics-15-00080-f005:**
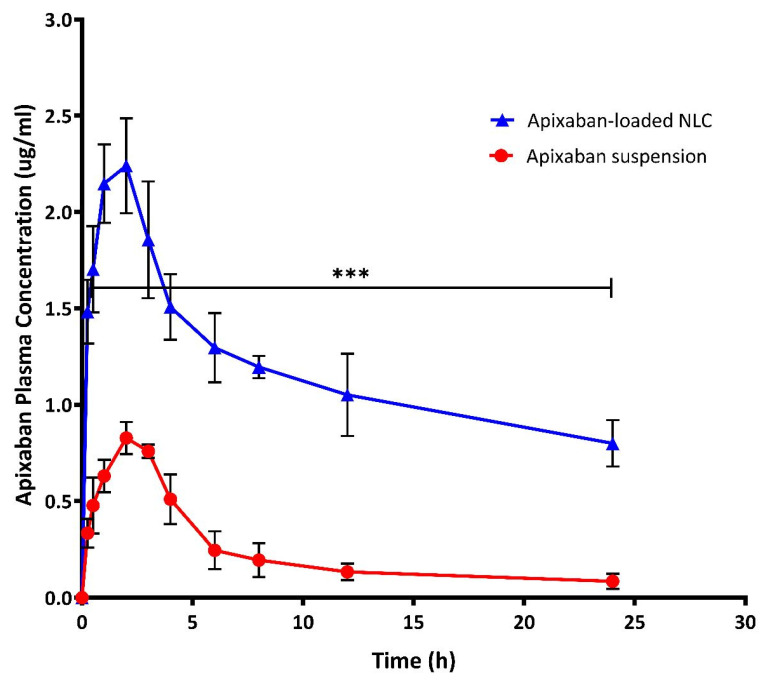
Plasma concentration of apixaban (Apx) after oral administration of 60 mg/kg of Apx- NLC and free Apx suspension (mean ± SD, *n* = 6). Note: *** significant effect at *p*-value < 0.001 at all-time points.

**Figure 6 pharmaceutics-15-00080-f006:**
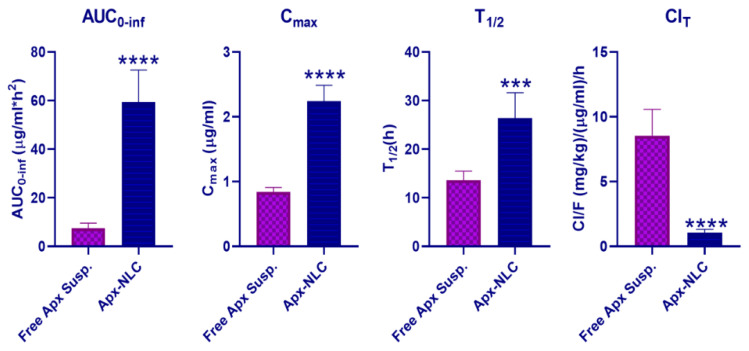
Statistical analysis of main pharmacokinetic parameters after oral administration of 60 mg/kg of both Apx- NLC and free Apx suspension (mean ± SD, *n* = 6). Note: *** and **** denote significant effects at *p*-values less than 0.001, and 0.0001, respectively.

**Figure 7 pharmaceutics-15-00080-f007:**
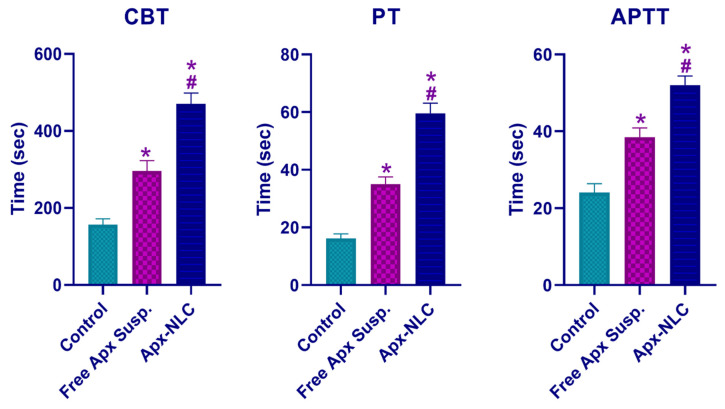
Cuticle bleeding time (CBT), prothrombin time (PT), and activated partial thromboplastin time (APTT) after oral administration of 60 mg/kg of both Apx- NLC and free Apx suspension compared with the control group (mean ± SD, *n* = 3). Note: * and # denote significant effects at *p*-values less than 0.05, with the free Apx suspension and control group, respectively.

**Table 1 pharmaceutics-15-00080-t001:** Independent and dependent variables of Box–Behnken design for the development of Apx-loaded NLC formulations.

Independent Variables (Factors)	Levels			Units
	Low (−1)	Medium (0)	High (+1)	
X_1_: Stearic acid amount	100	200	300	mg
X_2_: Tween 80 amount	100	200	300	mg
X_3_: Lecithin amount	0	50	100	mg
Dependent variables (Responses)	Units		Goal	
Y_1_: Mean particle size	nm		Minimize	
Y_2_: Zeta potential	mV		Maximize	
Y_3_: Entrapment efficiency	%		Maximize	

**Table 2 pharmaceutics-15-00080-t002:** Experimental runs and observed values of responses for BBD.

Runs	Factors			Responses		
	X_1_	X_2_	X_3_	Y_1_	Y_2_	Y_3_
	(mg)	(mg)	(mg)	(nm)	(mV)	(%)
F1	100	300	50	196.1	−41.2	79.4
F2	300	200	0	527.6	−24	42.5
F3	200	300	0	431.9	−23.4	50.5
F4	200	200	50	434.4	−42.3	76.3
F5	200	200	50	441.1	−42	75.3
F6	300	100	50	506.9	−42.6	72.4
F7	100	100	50	315.3	−44.6	75.2
F8	300	200	100	188.9	−46.8	76.5
F9	200	100	0	458.7	−26.8	48.2
F10	200	200	50	451.6	−42.4	75.3
F11	200	100	100	515.3	−44.1	82.7
F12	200	300	100	493.9	−41.8	81.8
F13	100	200	100	405.1	−47.5	94.1
F14	100	200	0	543.7	−25.9	57.2
F15	300	300	50	488	−38.8	72.4

Note: Oleic acid amount is 100 mg in all formulations.

**Table 3 pharmaceutics-15-00080-t003:** Estimated effects of factors associated with *p*-values for responses (Y_1_–Y_3_).

Factor	Y_1_		Y_2_		Y_3_	
	Factor Effect	*p*-Value	Factor Effect	*p*-Value	Factor Effect	*p*-Value
X_1_	62.8	0.0094 *	−1.75	0.0070 *	−10.525	0.0015 *
X_2_	−46.575	0.0169 *	−3.225	0.0021 *	1.4	0.0755
X_3_	−89.675	0.0046 *	20.025	0.0001 *	34.175	0.0001 *
X_1_^2^	−124.417	0.0052 *	1.592	0.0180 *	0.992	0.2407
X_1_X_2_	50.15	0.0286 *	−0.2	0.4380	−2.1	0.0680
X_1_X_3_	−100.05	0.0074 *	0.6	0.1022	−1.45	0.1286
X_2_^2^	−7.167	0.5103	−2.458	0.0077 *	−2.558	0.0510
X_2_X_3_	2.7	0.7849	0.55	0.1184	−1.6	0.1093
X_3_^2^	72.333	0.0152 *	−13.958	0.0002 *	−17.108	0.0012 *

Abbreviations: X_1_ is the Stearic acid amount (mg), X_2_ is the Tween 80 amount (mg), and X_3_ is the Lecithin amount (mg); X_1_X_2_, X_1_X_3_, and X_2_X_3_ are the interaction terms between the factors; X_1_^2^, X_2_^2^, and X_2_^3^ are the quadratic terms of the factors; Y_1_ is the mean vesicle size (nm), Y_2_ is the zeta potential and Y_3_ is the entrapment efficiency percentage; * Significant effect of factors on individual responses.

**Table 4 pharmaceutics-15-00080-t004:** Optimal calculated independent variables and observed, predicted, and residual values for dependent variables.

Independent Variables	Optimum (mg)	Dependent Variables	Predicted Values	Observed Values	Residuals	Prediction Error (%)
Stearic Acid amount (X_1_)	100	Mean particle size (Y_1_)	309.32 nm	315.2 nm	−5.88	1.9
Tween 80 amount (X_2_)	299.9	Zeta potential (Y_2_)	−44.43 mV	−43.4 mV	1.03	2.32
Lecithin amount (X_3_)	75.23	Entrapment efficiency (Y_3_)	88.27%	89.84%	−1.57	1.78

**Table 5 pharmaceutics-15-00080-t005:** Pharmacokinetic parameters of Apixaban-loaded NLC and free Apixaban suspension after single oral administration of 60 mg/kg to male Wistar rats, the data are expressed as mean ± SD (*n* = 6).

Parameter	Unit	Apixaban-Loaded NLC		Apixaban Suspension	
		Average	SD	Average	SD
Lambda_z	1/h	0.027 *	0.005	0.052	0.007
t_1/2_	H	26.398 *	5.23	13.6	1.89
Tmax	H	2	0	2.167	0.41
Cmax	μg/ml	2.2407 *	0.247	0.839	0.0699
AUC 0-t	μg/mL·h	28.369 *	3.92	5.75	1.353
AUC 0-inf	μg/mL·h	59.383 *	13.202	7.431	2.13
AUMC 0-inf	μg/mL·h^2^	2264.7 *	871.16	114.14	49.13
MRT 0-inf	H	37.01 *	6.775	14.916	2.279
Vz	(mg/kg)/(μg/mL)	38.819 *	3.38	167.257	42.748
Cl	(mg/kg)/(μg/mL)/h	1.0569 *	0.257	8.545	2.02

Note: * significantly different from values of free Apixaban suspension at *p*-value < 0.05.

## Data Availability

Not applicable.

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
