# Peer review of "Tailoring Apixaban in Nanostructured Lipid Carrier Enhancing Its Oral Bioavailability and Anticoagulant Activity"

_pharmaceutics, 2022, doi:10.3390/pharmaceutics15010080_

Round 1

Reviewer 1 Report (Previous Reviewer 1)

I appreciate the author's reply and revision. The author's explanation of the 9-fold increase in AUC is acceptable.

Author Response

We deeply thank the reviewer for his appreciation and support.

Reviewer 2 Report (New Reviewer)

Abstract:

What does the sentence “The prepared, characterized, morphologically studied, and pharmacokinetically and pharmacodynamically assessed optimal formula.” mean? Seems like there is something missing.

Keep numbers consistent. At times you have said “8 folds” and at other times you have said “eight folds”.

It is also unclear how half-life was only affected 2-fold while clearance was affected 8-fold. Half life is proportional to clearance, unless the authors claim that the particles have impacted the drug volume of distribution.

Introduction:

As with previous inconsistencies, the authors have written “The dose of Apx for the treatment of VTE is 2.5 mg twice daily up to 5 mg BID”. Please use either “twice daily” or “BID”, not both. Also, BID is not a widely understood abbreviation.

The Cmax and AUC are difficult to appreciate as low without any context. At present, the authors have just provided values but this does not really justify limitations of current formulations. Twice daily dosing seems like a reasonable regimen.

Materials and methods:

Table 2 should be in the results section. Table 1 is fine in methods. If both need to be back to back, then both should be moved to results.

For preparation of Apx-loaded NLCs, what was the chloroform to methanol ratio? Also was a cryoprotectant included if the formulation was frozen to -20 °C right after sonication? This freezing may rupture the NLC shell.

For separation and washing of Apx-loaded NLCs, what specific temperature was used to thaw the NLCs? The only value shown is -4 °C and this is definitely below the assumed preparation temperature of 60 °C.

For EE% determination, the micropipette currently says “100-1000 L”. A micro sign appears to be missing. How were the particles “burst”? This isn’t clear.

For in vitro release, was the PBS with 0.05% SLS added TO the dialysis bags or to the external release media? The current wording makes it sound like it was put into the dialysis bags.

For TEM, it isn’t clear what “staining solution” refers to.

For PT and APTT, it isn’t clear how animals are “treated to produce plasma”.

Results and discussion:

What is the ± of the %EE? The mean %EE’s do not look that different, yet the authors claim a p<0.05. It is very surprising to see the %EE being so consistent.

For the “variables” results paragraph, there is a lot of text that is difficult to follow. The authors are asked to break sections down into smaller, easier to follow paragraphs. A figure or table should also be referred to making it easier for the reader to understand the trends being described.

Table 3 – the table heading should be extended to include definitions of the different X and Y variables.

Figure 1 – What are the units for the numbers on the x-axis? E.g. stearic acid “100 to 300”

The text before Table 3 is cut short and then continued after Figure 2. It would make more sense for the text to be displayed in a continuous manner.

Please keep referring to figures and tables. When reading through the “Effects on the Mean Particle Size (Y1)” text, it was difficult to know what specific data was being discussed. In particular, the notation for formulations (F13, F14, etc.) was difficult to interpret. Also a continued remark that breaking the text into smaller paragraphs would make this manuscript a lot easier to follow.

There is some repetition e.g. Tween 80 reducing the zeta potential magnitude has been mentioned twice in the results/discussion.

Equations 2, 3 and 4 need to be made more presentable. They are currently very difficult to understand.

The lipids are being presented as weights, but would it not be more valuable to present them as concentrations? Will the formulation be administered in solution or as a solid? In any case, reporting the variables as “100 mg, 299.9 mg, and 75.23 mg” is confusing to the reader beyond just being a ratio of the lipids.

Table 4 appears to have alignment issues.

Figure 4a needs error bars. The legend indicates that these are present, but they cannot be seen.

Figure 4b – are sample images of more than a singular particle available?

Author Response

This manuscript is a resubmission of an earlier submission. The following is a list of the peer review reports and author responses from that submission.

Round 1

Reviewer 1 Report

The authors did a relatively complete study of the apixaban-loaded nanostructured lipid carrier (APX-NLC), except for the lack of stability studies. In addition, the following suggestions are for the author's reference.

(1) Regarding the significance of the study: As the authors stated, 'The dose of Apx for the treatment of VTE is 2.5 mg twice daily up to 5 mg BID', and the oral bioavailability of apixaban's marketed tablet is over 50%, which was not low, especially given its small daily oral dose. Therefore, is it meaningful to study the nano oral formulation of apixaban, especially for the pharmaceutical industry? It is strongly recommended that the meaning of the research be clearly stated in the Introduction.

(2) Pharmacokinetic results: The authors reported that the AUC and Cmax of APX-NLC were 9-fold and 2.7-fold higher than that of the suspension, respectively, which is very surprising and probably also controversial, especially considering that the oral bioavailability of APX had exceeded 50%. Therefore, it is recommended to explain the reasons in depth in the discussion section and supplement the pharmacokinetic data of APX's marketed oral tablets and injections as reference preparations.

Author Response

Kindly, Please see the attachment.

Reviewer 2 Report

In the manuscript by Zaky et al, “Tailoring Apixaban in Nanostructured Lipid Carrier Enhancing Its Oral Bioavailability and Anticoagulant Activity”. In this manuscript authors tried to enhance the oral bioavailability of Apixaban which is an oral anticoagulant. Authors have tried to optimize the formulation by using Box-behnken design and optimized formulation (nanostructured lipid carrier) was characterized and in vivo pharmacokinetic study was conducted in male wister rats. This is interesting manuscript with well-designed experiments. However, authors have to address the following comments/suggestions before publication.

- In introduction, authors mentioned that the oral bioavailability of Apx is 50% reported, I think this not too low, so cross check with literature report.

-Authors have well explained about SLNs and NLCs in introduction, it is also recommended to few examples from literature support that how these NLCs enhanced the oral bioavailability of poor aqueous soluble drugs. These may be bypassing the first pass metabolism (promotes oral absorption through lymphatic transport) and lipid digestion and formation of micelles or mixed micelles as explained in below manuscript.

-https://doi.org/10.1016%2Fj.jsps.2021.07.015 

-https://doi.org/10.1016/j.ijpharm.2014.11.001

-Degree Celsius writing format is not correct, need to correct.

-Formulation design contents equal amount of Tween 80 and Stearic acid, Tween 80 being nonionic surfactant it should not be such high amount in the formulation, check the limit of tween 80 permitted to use.

-Ratio of liquid lipid to solid lipid should not be more than 1:3, formulation needs optimization more.

-There are a lot of grammatical mistakes in the manuscript, it is recommended to take advise from English language center or native English speaker.

- PDI of the formulations range from 0.35 – 0.57, looks higher PDI indicating heterogeneity in particle size distributions. What could be the reason?

-Figure captions should include data presented as and with number of replicates. Error bars look very small, do authors properly put the correct error bars?

-  Figure 5, cross check again, at 0 hr time point what is plasma concentration, it seems that 0 hr is not 0.

-Error bars for figure 5 same, how did authors obtain so low error bar on animal experiments? Double check the data again.

Reviewer 3 Report

Dear editors and authors:

The authors developed the apixaban-loaded nanostructured lipid carrier formula to enhance the bioavailability of apixaban. However, the experiment design was poor, and the results were not reliable. Based on the criteria of the Journal of metabolites, I decided to reject the manuscript due to its lack of novelty and reliability.

The major concerns:

1.     The authors results were not correct and can’t increase in AUC and Cmax by 9.26 and 2.7 folds, respectively, compared to oral apixaban suspension. Because the absolute oral bioavailability of apixaban-loaded nanostructured lipid carrier formula can’t over 100%. As we known that apixaban is well absorbed in rat, dog, chimpanzee and human, with absolute oral bioavailability of proximately 50% or greater, if the apixaban-loaded nanostructured lipid carrier formula can increase the AUC by 9.26-fold, the absolute oral bioavailability of apixaban-loaded nanostructured lipid carrier formula was 500%?secondly, I didn’t think the bioavailability is poor for apixaban.

2.     The experiment design was poor. Patients dosage of apixaban is 5 or 10 mg BID(70KG), it means 0.15mg/kg, equal to rat dosage =0.15mg/kg*6.3(transform factors) =0.9mg/kg. The authors used 60mg/kg in rodent study, was too high.

Round 2

Reviewer 1 Report

Many thanks to the authors for the earnest and quick reply. Although the authors offer some explanations and cite literature, in my opinion, the significance of this study is still not fully demonstrated, especially the reason for the 9-fold increase in AUC is not clearly explained. I would like to suggest that the authors consider the possibility that the rats were overdosed beyond linear pharmacokinetics. Maybe more evidence is needed to explain it.

Reviewer 2 Report

The pharmacokinetic data presented by the authors look very low SD, which is not at all possible in animal experiement.

Reviewer 3 Report

Dear editors and authors:

The manuscript hasn't been improved. Actually, the authors didn't directly answer my questions. Therefore, I insist on rejecting the manuscript due to its lack of novelty and reliability. The authors need to discuss the possible reasons and persuade readers to agree with their results.

e.g. the authors cited other researchers’ data to explain that the apixaban-NCL formulation can enhance apixaban's AUC by 9.6-fold. But a cocrystal of apixaban with oxalic acid improved the solubility and consequently the oral bioavailability of apixaban by 2-fold via a cocrystal. Their mechanism is different from your study, what’s your mechanism?  The authors need to explain the possible reasons not only citing the results of other studies.

The same thing, the authors can’t say that the we used published dosage of apixaban even the dosage is too high.